# Endothelin-1 in Health and Disease

**DOI:** 10.3390/ijms241411295

**Published:** 2023-07-10

**Authors:** Katherine M. R. M. Banecki, Kim A. Dora

**Affiliations:** Department of Pharmacology, University of Oxford, Mansfield Road, Oxford OX1 3QT, UK; katherine.banecki@st-hildas.ox.ac.uk

**Keywords:** endothelin-1, storage, release, vasculature, endothelium, cardiovascular disease

## Abstract

Discovered almost 40 years ago, the potent vasoconstrictor peptide endothelin-1 (ET-1) has a wide range of roles both physiologically and pathologically. In recent years, there has been a focus on the contribution of ET-1 to disease. This has led to the development of various ET receptor antagonists, some of which are approved for the treatment of pulmonary arterial hypertension, while clinical trials for other diseases have been numerous yet, for the most part, unsuccessful. However, given the vast physiological impact of ET-1, it is both surprising and disappointing that therapeutics targeting the ET-1 pathway remain limited. Strategies aimed at the pathways influencing the synthesis and release of ET-1 could provide new therapeutic avenues, yet research using cultured cells in vitro has had little follow up in intact ex vivo and in vivo preparations. This article summarises what is currently known about the synthesis, storage and release of ET-1 as well as the role of ET-1 in several diseases including cardiovascular diseases, COVID-19 and chronic pain. Unravelling the ET-1 pathway and identifying therapeutic targets has the potential to treat many diseases whether through disease prevention, slowing disease progression or reversing pathology.

## 1. Introduction

First isolated from the supernatant of porcine aortic endothelial cells (ECs) in 1988 [1], endothelin-1 (ET-1) is a 21 amino acid potent vasoconstrictor and mitogen. In the vasculature, ET-1 is important for vasoconstriction and the control of vascular tone [2] and, throughout the body, it has functions such as the control of ion transport in the gastrointestinal tract and the recruitment of immune cells in inflammation.

In the years following the identification of ET-1, a wave of publications investigated how ET-1 was synthesised, stored and released. However, much of this work was performed using cultured cells of various species in vitro, with remarkably few of these initial findings translated to intact tissues ex vivo and/or whole organisms in vivo (Table 1). Furthermore, studies of the vasculature have lacked clear distinctions between vascular beds and the macro-cf. microvasculature. Therefore, by focusing on cultured cells, our understanding may not truly represent the various physiological situations found throughout the body in vivo. Cultured cells may be more closely associated with a state of “injury” due to the adaption of the cells to an artificial environment outside of their in vivo cellular niches [3], and therefore might be more applicable to the ET-1 pathway in pathology. Another consideration is the use of human umbilical vein ECs (HUVECs), where conclusions should perhaps be interpreted with caution due to potential differences between arterial and venous ECs [4]. Therefore, it is clear that work is still required to translate findings to isolated intact arteries. This may be timely with improved imaging and molecular techniques, particularly in recent years. Overall, there is a substantial gap in the literature leaving many unanswered questions as to the synthesis, storage and release of ET-1 in health and disease.

## 2. ET-1

### 2.1. ET-1 Synthesis

As the name suggests, endothelin-1 was first identified in ECs [1], however, the production of ET-1 has now been reported in many cell types. For example, ET-1 has been found to be expressed basally in pulmonary epithelial cells [63] and keratinocytes [64]. There is also evidence for the stimulated release of ET-1 from a variety of immune cells, such as macrophages [65,66,67] and dendritic cells [68], suggesting synthesis in those cells.

ET-1 is transcribed from the gene, *Edn1*, present on chromosome 6. The gene is transcribed, forming the 2.8-kb messenger RNA (mRNA), and is translated into a 212 amino acid precursor protein, preproET-1. Mature ET-1 is encoded in exon 2 of the 5 exon 6.8 kb preproET-1 gene. The transcription of *Edn1* can be regulated by numerous transcription factors such as hypoxia inducible factor 1 (HIF-1) and activator protein 1 (AP-1) [69], allowing the control of ET-1 formation by many different physiological and pathophysiological factors in different cells and tissues. *Edn1* gene transcriptional control is covered extensively elsewhere [69]. It is also possible to control ET-1 expression through epigenetic regulation and mRNA stability [69].

Once translated, preproET-1 is then processed via a series of cleavage steps to generate the mature peptide. PreproET-1 is processed by a furin-type proprotein convertase to the inactive intermediate, big ET-1 (38 amino acids). Big ET-1 can then be cleaved by the endothelin-converting enzyme (ECE) to produce the mature 21 amino acid peptide, ET-1 [7]. A two-step process involving chymase may also generate ET-1 (1–31) from big ET-1; this can then be cleaved by neprilysin to generate mature ET-1 [8,9].

### 2.2. ET-1 Function

When released, ET-1 can act via the G protein-coupled receptor (GPCR) subtypes known as ET_A_ [70] and ET_B_ [71]. These receptors are present through the body representing the multi-functionality of ET-1. There is evidence to suggest G-protein promiscuity of ET_A_ and ET_B_ receptors, meaning they can interact with G_q_, G_i_, G_12/13_ and G_s_ to enact different cellular effects [45,47,72]. These receptors can also be coupled to β-arrestin [46,48]. This may be important for the downregulation of G protein signalling via the internalisation of ET receptors for recycling or degradation [73]. There is also evidence that ET-1 may be able to reenforce and prolong signalling through the β-arrestin pathway [74]. The multitude of G protein signalling pathways that ET-1 may activate explains how ET-1 has so many functions in the body beyond simply controlling blood flow and muscle contraction, for example, ion transport, vascular permeability and inflammation.

#### 2.2.1. ET-1 in the Vasculature

In the vasculature, ET-1 influences blood pressure via vasoconstriction and is important in the control of basal vascular tone [2]. In human participants, the administration of the ET_A/B_ receptor antagonist, TAK-044, resulted in continuous, dose-dependent peripheral vasodilation and hypotension, as well as a rapid, dose-dependent increase in plasma ET-1, suggesting hypotension may trigger ET-1 release [49]. This antagonist also abolished the local forearm vasoconstriction observed upon arterial infusion of ET-1, again demonstrating the importance of ET-1 in regulating basal vascular tone [49].

However, in vivo haemodynamic responses to intravenously injected ET-1 are complex and depend on the vascular bed, in part due to differences in ET-1 receptor expression [75]. The effects of endogenous ET-1 may also differ from infused doses as it appears that the majority of ET-1 released by (cultured) ECs would be in the direction of the SMCs, i.e., basolaterally, facilitating the ability of ET-1 to act as a local paracrine regulator of vascular tone [27,76]. The polarity of secretion has also been demonstrated in non-vascular epithelial cells, with 30 times more ET-1 released towards the submucosal side in the airways compared with the apical side [30].

Vascular responses to ET-1 include both direct vasoconstriction, as well as indirect responses such as stimulating the growth of vascular smooth muscle cells (VSMCs) [77] (Figure 1). ET_A_ receptors are predominantly expressed by VSMCs. In contrast, while ET_B_ receptors can also be expressed in VSMCs, they are thought to be predominately expressed by ECs, suggesting ET-1 can act in an autocrine manner [78]. These receptors are G_q_-coupled, resulting in VSMC contraction via increased intracellular calcium, and EC-dependent vasodilation via the release of nitric oxide (NO) and prostaglandin I_2_ [79]. NO can also inhibit further ET-1 release [75]. Furthermore, ET_B_ receptors may also act as clearance receptors to rapidly remove ET-1 from circulation [77,78], although the evidence for this mechanism is unclear.

#### 2.2.2. ET-1 in the Heart

Although ET-1 can have indirect effects on cardiac muscle by modulating coronary artery tone, it can also have direct effects on the muscle and affect cardiac output. In volunteers with atypical chest pain (i.e., normal left ventricular function and epicardial coronary arteries), there was a significant reduction in left ventricular dP/dt following left coronary artery infusion of ET_A_ receptor antagonist, BQ-123, suggesting a positive and ongoing inotropic role for endogenous ET-1 [56].

This was explored further using isolated human cardiac tissue, where differences in responses to ET-1 were observed across chambers. In both the left ventricle and right atrium, ET-1 caused an increase in contractile force, however, in the right atrium, this was preceded by an initial transient reduction in contraction [55]. The difference was attributed to the expression of ET_B_ receptors in the atrium, as the ET_B_ receptor antagonist, Ro468443, prevented the initial ET-1-mediated decrease in contractility. When ET-1 was applied to left ventricular myocytes isolated from human cardiac biopsies, there was an observed increase in myocyte shortening which could be completely prevented by BQ-123 [54], demonstrating ET-1 has a positive inotropic effect, mediated by ET_A_ receptors.

#### 2.2.3. ET-1 in the Airways

As with the vasculature, ET-1 stimulates the contraction of SMCs in the airways. However, unlike the vasculature, this effect appears to be mediated by ET_B_ receptors as bronchoconstriction is not inhibited by ET_A_ antagonists [60]. Bronchoconstriction is important in airway defense as it protects the body against inhaled irritants and toxins.

ET-1 also has effects on epithelial ion transport such as chloride secretion, regulating the volume of airway surface secretions and fluids. In human bronchial epithelial cells mounted in an Ussing chamber, ET-1 stimulated Cl^−^ secretion, leading to fluid secretion [59]. Again, this effect appeared to be due to ET_B_ receptors as BQ-788, an ET_B_ receptor antagonist, but not BQ-123, prevented the effect of ET-1.

#### 2.2.4. ET-1 in the Gastrointestinal Tract

ET-1-induced SMC contraction is also important in the gastrointestinal tract. ET-1 applied to strips of rat stomach, colon and guinea pig ileum leads to contraction [61], suggesting the role of ET-1 in peristalsis.

Similar to the effect of ET-1 on the airways, it appears to have a role in absorption and secretion in the gut. In isolated muscle-stripped human intestinal mucosa in an Ussing chamber, ET-1 was again shown to stimulate Cl^−^ secretion but it was also shown to inhibit sodium and glucose absorption through the Na^+^-glucose cotransporter [62]. The modulation of ion transport in the gut is important for the regulation of water, solute and electrolyte homeostasis.

#### 2.2.5. ET-1 in the Kidneys

In the kidneys, ET-1 causes the contraction of mesangial cells in a similar manner to that of VSMCs [57], thus allowing the modulation of the glomerular filtration rate.

ET-1 has also been shown to modulate fluid and electrolyte reabsorption in the kidney, independently of alterations in the glomerular filtration rate. In the cortical collecting tubule, ET-1 can inhibit sodium and water reabsorption; the mechanism behind this has been demonstrated in isolated collecting tubules where, via the activation of PKC, ET-1 reduced the arginine vasopressin-mediated cAMP accumulation required for insertion of aquaporins in the apical membrane [58].

#### 2.2.6. ET-1 in Inflammation

ET-1’s role in the immune response has led to it being labelled as a “profibrotic cytokine” in itself [80].

ET-1 can modulate vascular permeability, which is important in the immune response. In conscious rats administered with ET-1, a dose-dependent increase in haematocrit was observed [52]. Evans blue dye was used to assess vascular permeability and increased extravasation was demonstrated in several vascular beds including the airways, gastrointestinal tract and kidney. The same approach showed vascular permeability was increased by ET-1 in the rat heart, interestingly through the ET_A_ receptor [53], which may link to the EC expression of this receptor in the coronary vasculature.

ET-1 also influences the immune response via other mechanisms, including the transcription of proinflammatory cytokines. ET-1 increases the expression of cytokines, such as tumour necrosis factor-α (TNF-α), interleukin-1 (IL-1) and interleukin-6 (IL-6), from monocytes [50]. These cytokines can promote chemotaxis, a process ET-1 can also stimulate [51]. Importantly, this can result in a positive feedback system, as these cytokines also stimulate ET-1 synthesis and release [37]. Therefore, ET receptor antagonists could provide the therapeutic potential to reduce inflammatory responses.

So, although ET-1 was initially identified in the vasculature, the effects of ET-1 go far beyond blood pressure control and appear relevant in essentially all major organs. Alterations in ET-1 levels, therefore, can be predicted to have body-wide implications in pathology and this may explain why ET-1 has been associated with a great number of diseases.

## 3. Intracellular Storage of ET-1

### 3.1. In Vitro

It is generally agreed that ET-1 can be stored in cytoplasmic secretory (exocytic) vesicles in ECs; yet the literature is inconsistent regarding the storage organelle(s) for ET-1 in ECs (Table 1). For example, ET-1 has been reported to be stored only in secretory vesicles [14,15], Weibel–Palade bodies (WPBs) [16,17,20] or not stored at all and merely present in the cytoplasm [22,81]. There is likely to be some cross-over in these characterisations, but certain organelles can be defined, such as the WPBs (by electron microscopy, or fluorescence microscopy for immunolabelled von Willebrand factor (vWF) [82]). Most of these studies used antibodies against ET-1, which would also detect its precursors such as prepro-ET-1, so the term ‘ET-1-like’ immunolabel has often been used.

Cycloheximide, a protein synthesis inhibitor, can be used as functional evidence for or against ET-1 storage. Treating coronary microvascular ECs from the adult rat ventricle with cycloheximide had no effect on the release of ET-1 and, therefore, any ET-1 released during this time must be from a preformed intracellular pool, i.e., storage [10]. In contrast, in porcine aortic ECs, both the basal and thrombin-induced release of ET-1 could be inhibited in a dose-dependent manner by cycloheximide, suggesting on-demand protein synthesis [12]. One explanation is that this could be due to differences between resistance arteries and larger vessels [83].

The most cited evidence regarding ET-1 storage in WPBs has come from cultured cells. Russell et al. reported the predominant localisation of ET immunoreactivity to the cytosol (including secretory vesicles), and also to WPBs, in human coronary artery ECs [16]. They also reported some immunolabelling in vesicles budding from the plasma membrane, corresponding to caveolin labelling, demonstrating ET-1 within the endocytic pathway; this could merely demonstrate the recycling of ET-1 but it is also potentially a mechanism by which ECs can prolong cellular responses to ET-1 via peptide recycling [84] (Figure 2). The potential storage of ET-1 within the endocytic pathway is something that has not been studied further.

WPBs have also been shown to store pro-ET-1 and ECE-1 [17], implying this may also be the site of ET-1 synthesis and/or conversion. On the isolation of exocytic vesicles from cultured bovine aortic ECs, Harrison et al. reported the presence of ET-1, but not ECE, suggesting a non-vesicular role for ECE [15]. Due to the differential localisation of ECE isoforms [1], it is likely that ECE is important in both the vesicular and cytoplasmic processing of ET-1, as well as having an extracellular role.

In bovine pulmonary ECs, it has been demonstrated that ET-1 is present diffusely within the cytoplasm, and was not localised to cytoplasmic vesicles or WPBs [21], suggesting the release of ET-1 via a mechanism that does not require prior storage. ET-1-like immunolabel was reported on the rough endoplasmic reticulum, Golgi cistern, small vesicles positioned beneath the cell membrane and lysosomes, but no ‘classic’ electron-dense secretory granules were found [21].

### 3.2. Ex Vivo and In Vivo

Rudimentary in vivo evidence of ET-1 storage comes from the rapid temporal dynamics of its release. During a cold pressor test in humans, which involves the immersion of one forearm in ice-water (resulting in vasoconstriction), the other arm acting as a control, ET-1 release was detected in 2 min in the submerged arm, and in the control arm after a short delay [11]. Both diastolic and systolic blood pressure increased following the increase in plasma ET-1. This suggests a relatively fast, localised release of ET-1 from the submerged forearm which leads to a global increase in plasma ET-1 which could at least contribute to the increase in blood pressure. The mechanism for ET-1 release by this in vivo test is still not known, but one possibility is that the cold activates a TRP channel (e.g., TRPA1) [85,86] to signal ET-1 release from vascular ECs.

Moving ex vivo, evidence against the storage of ET-1 in porcine thoracic aortic strips was demonstrated using cycloheximide [13]. Both the basal and thrombin-induced release of ET-1 could be prevented by cycloheximide, suggesting ET-1 is synthesised on demand. When the endothelium was removed, there was no detectable release of ET-1 or big ET-1, demonstrating the origin of ET-1 to be the ECs.

Similar to the in vitro studies, a homogeneous pattern of staining between ET-1 and vWF has been observed in endocardial ECs as well as in larger coronary and microvascular ECs, suggesting the WPBs are the storage site for ET-1 in the heart [18]. They also reported ET-1 in the cardiomyocytes. This study used endomyocardial biopsies from heart transplant patients where ET-1 levels correlated with areas of ischaemia, fibrosis and damage, thus, potentially more closely representing a pathological context. This area clearly warrants further investigation.

ET-1 storage in WPBs has also been shown in blood vessels of the carotid body in rats [20]. ET-1 has also been localised to WPBs in cadmium sulphate-treated rats [19]. Cadmium sulphate causes arterial injury and can lead to hypertension. This treatment was found to increase the number of WPBs, their degranulation and exocytosis, and the release of ET-1 from aortic ECs. These interesting observations suggest a functional link in vivo [19], whether blood pressure and inflammatory responses were increased and sensitive to ET receptor block requires further investigation.

Not all the literature supports the WPBs as the storage organelle for ET-1 in ECs. In ECs of rat aortic rings, ET-1 immunofluorescence rarely aligned with vWF [14], suggesting that the primary storage site in ECs is exocytic vesicles. Interestingly, ET-1 was not present basally in aortic rings of young or old rats. ET-1 expression could only be stimulated in old rats after 5 min of exposure to thrombin, which caused a rapid, 3.7-fold increase in ET-1 expression, a process, therefore, linked to vascular aging. The rapid formation was proposed to be possible if ET-1 was stored as a precursor and formed during the exocytotic process. However, the antibody used would also bind precursors of ET-1, so it is unlikely there is basal storage of a precursor either, meaning future investigations are required.

In skin biopsies from Type 1 diabetic patients ET-1-like immunolabel was found diffusely within the cytoplasm, not localised to WPBs or any cytoplasmic vesicles [22]. However, due to the nature of the patients, this may represent a more pathological state.

In summary, if ECs make *Edn1* mRNA basally or in response to stimuli, the peptide can be made in ECs. Where and how it is stored, as mRNA or peptide, may vary between vascular beds, tissues and species, potentially explaining some of the discrepancies in the results reviewed here.

## 4. Release of ET-1

Given the numerous cell types known to express ET-1, it is possible that many different release pathways may be important, depending on the cell type, vascular bed and species. It has been proposed that ET-1 can be released by a more regulated pathway via the WPBs in response to agonists and a constitutive release pathway via secretory vesicles (Figure 2). Furthermore, given the many different stimuli for ET-1 release, each of these may activate one of several different proposed pathways which enables the differential release of ET-1 under different pathophysiologic conditions. Exactly how these factors cause the appropriate response, whether that be a stimulation or inhibition of ET-1 release, has also been investigated, for the most part, in cultured cells (Table 1).

It is likely that intracellular calcium, from both external and internal sources, is important for ET-1 release [12,17,25,26], particularly as stimuli of ET-1 release, such as thrombin [12] and angiotensin II [87], cause an increase in intracellular calcium. If stored in WPBs, their release may also require calcium to activate exocytosis [88]. However, how cAMP plays a role is less clear, as cAMP has been shown to stimulate WPB exocytosis [88], but inhibit ET-1 release [31,32,33,89].

### 4.1. Thrombin-Induced ET-1 Release

Thrombin is a potent stimulator of ET-1 release and likely causes release via the thrombin-induced activation of PKC [12,13], intracellular calcium mobilisation [12], and/or the activation of the ET-1 gene promoter [69], events which can act synergistically.

#### 4.1.1. In Vitro

The transcriptional activation of the *Edn1* gene by thrombin has been explored (Table 1) and underlies an effect to cause ET-1 release from ECs. Thrombin activates the *Edn1* promoter via the AP-1 binding site at position −108 bp [69]. The AP-1 binding site is one of the most extensively studied of the *Edn1* gene. AP-1 proteins generally require the formation of homodimers or heterodimers between members of the Jun and Fos families [90]. AP-1 is involved in genetic responses to growth factors and proinflammatory signals as well as the basal regulation of *Edn1* transcription in bovine aortic ECs [91].

Thrombin can promote ET-1 gene expression via both the PKC-dependent [5] and -independent [29] activation of AP-1. In porcine aortic ECs, a thrombin-induced increase in preproET-1 mRNA (detected via northern blotting) was completely inhibited by a PKC inhibitor, calphostin C [5]. It was suggested this was due to thrombin-induced synthesis and dephosphorylation of c-Jun which occurs within 5 min of thrombin exposure; c-Jun could then go on to bind to the AP-1 binding site of the *Edn1* promoter and induce ET-1 expression. However, in a separate study using a variety of macrovascular and microvascular ECs (HUVECs, bovine pulmonary artery ECs and a human microvascular EC line), the thrombin-stimulated increase in preproET-1 mRNA was neither suppressed by the PKC inhibitors sangivamycin and calphostin C, nor by PKC depletion. Instead, they proposed a role for protein tyrosine kinases (PTK) as they found thrombin stimulated an increase in phosphotyrosine-containing proteins and inhibitors of PTK (herbimycin A and genistein) could block the thrombin-stimulated increase in preproET-1 mRNA and subsequent peptide secretion [29].

The role of intracellular calcium has been proposed to be important for triggering the thrombin-induced release of ET-1 from porcine aortic ECs [26]. Intracellular calcium acting via the Ca^2+^-calmodulin complex results in the phosphorylation of myosin light chain (MLC) by MLC kinase which facilitates the formation of filamentous actin and activated myosin. It is possible MLC facilitates ET-1 release via the transport of ET-1-containing vesicles to the cell membrane. Intracellular calcium is also likely to be responsible for ET-1 release in response to angiotensin II and vasopressin as both induce the receptor-mediated breakdown of phosphoinositide via phospholipase C [87].

Furthermore, low shear stress has also been shown to increase ET-1 release [28] as well as intracellular free Ca^2+^ concentrations [92]. Russell et al. found that, when treating HUVECs with a calcium ionophore at a very high concentration, ET-1 was released [17]. This further suggests the release of ET-1 via a regulated secretory pathway involving calcium (Figure 2).

#### 4.1.2. Ex Vivo and In Vivo

The thrombin-induced release of ET-1 has been demonstrated in the aortic rings of old rats [14]. Exposure to thrombin for 5 min caused an increase in ET-1 peptide expression, as well as an increase in the expression of big ET-1, preproET-1 and ECE-1 mRNA, suggesting the upregulation of transcription, although the mechanisms of upregulation of transcription and stimulation of release were not investigated directly.

### 4.2. Regulation of ET-1 Release by Shear Stress

#### In Vitro

The effects of shear stress on ET-1 release are potentially very important, as ECs are constantly exposed to varying degrees of shear stress. With cell culture studies, shear stress is not usually applied to the cells and it is important to take this into consideration.

Physiological levels of shear stress inhibit ET-1 transcription and release. In bovine aortic ECs exposed to steady laminar fluid, a shear stress of 15 dyn/cm^2^ and a 4–5-fold decrease in ET-1 mRNA, as measured by northern blotting, was observed. This commenced in as little as 1 h and reached completion within 2 h. This was also dependent on the magnitude of shear stress with cells exposed to <15 dyn/cm^2^ having greater levels of ET-1 mRNA. ET-1 peptide levels at 24 and 48 h reflected the changes seen in mRNA as the levels of immunoreactive ET-1 in the culture media also decreased [39]. The precise mechanisms were not explored.

The inhibition of ET-1 release by high shear stress of up to 25 dyn/cm^2^ has been shown in HUVECs [28]. This inhibition could be partially reversed with a pre-treatment of N(G)-nitro-L-arginine (L-NNA) and a NOS inhibitor, and was mimicked by 8-bromo-cyclic GMP. L-NNA appeared to reduce intracellular cyclic GMP (cGMP) by 40%, however, it was only with the addition of methylene blue that the decrease in ET-1 release could be fully reversed. While this was not explained, it does implicate NO and the subsequent accumulation of cGMP in the observed decrease in ET-1 release during exposure to high shear stress. Therefore, NO acts both to balance ET-1 release and contraction, and flow is a mediator of ET-1 release. When HUVECs were exposed to low level shear stress or a brief exposure to higher level shear stress, ET-1 release was stimulated. A PKC-dependent mechanism has been identified as staurosporine (PKC inhibitor) completely blocked this stimulated release of ET-1. How this links to PTKs has not been explored (see Section 4.1.1). Nevertheless, the observation that differing exposure times and flow rates may have opposing effects on ET-1 release may help explain inconsistences in the literature.

Due to difficulties in monitoring shear stress in vivo, there is a lack of investigations of the effects of shear stress in whole organisms; and this has yet to be explored in isolated, pressurised arteries with measurements of luminal shear stress.

### 4.3. cAMP Regulation of ET-1 Release

#### 4.3.1. In Vitro

cAMP may be an important regulator of ET-1 release via adenosine. For example, adenosine acts via A_2_ receptors to inhibit basal ET-1 release from guinea pig tracheal epithelial cells. An increase in intracellular cAMP was accompanied by a decrease in ET-1 production [31]. Similarly, in guinea pig tracheal epithelial cells, the inhibition of both basal and LPS-stimulated ET-1 release by cAMP has been reported. 8-bromo-cyclic AMP and forskolin, which were confirmed to increase cAMP levels intracellularly, reduced the basal and stimulated ET-1 release [32].

The cytokine-induced (TNF-α and interferon-γ (IFN-γ)) release of ET-1 from human VSMCs can also be inhibited by cAMP [89]. A prostacyclin analogue, forskolin and a phosphodiesterase type IV inhibitor inhibited ET-1 release in a concentration-dependent manner. Furthermore, phosphodiesterase type IV inhibition also inhibited the cytokine-induced upregulation of preproET-1 mRNA expression in VSMCs, suggesting cAMP also acts to inhibit ET-1 at the level of transcription. Nevertheless, it is not clear if this represents normal physiology, as these cells were isolated from patients undergoing coronary artery bypass surgery.

In the heart, cAMP may also be important for ET-1 regulation [33]. Neuropeptide Y stimulated ET-1 release from human endocardial ECs, the mechanism of which involves both Y2 and Y5 receptors. These receptors are G_i_-coupled and, therefore, reduce intracellular cAMP, thus relieving the inhibition on ET-1 release. The role of cAMP within ECs of the vasculature has not been investigated.

#### 4.3.2. Ex Vivo and In Vivo

The mechanism by which cAMP may inhibit ET-1 release has been investigated in a Langendorff-perfused rat heart preparation [38]. When treating hearts and primary human coronary artery ECs with a K_ATP_ channel blocker, tolbutamide, an increase in ET-1 levels in the perfusate and culture media, respectively, compared to the control, was observed after 20 min. This evidence was also accompanied by immunocytochemistry, which demonstrated a similar expression pattern for K_ATP_ channels and ET-1 in ECs. This could suggest a mechanism by which cAMP activates the K_ATP_ channel in close proximity to ET-1 storage vesicles, reducing ET-1 release. How the signalling is linked should be investigated further.

None of these cited studies established whether WPB (vWF) and ET-1 release aligned, so, given the opposing effect cAMP seems to have on ET-1 and WPB release, this requires further investigation.

## 5. Effect of NO on Actions of ET-1

NO and ET-1 are frequently referred to as the balance necessary to sustain a ‘healthy’ endothelium. This is due to the ability of NO to antagonise the formation and actions of ET-1. The concentrations at which NO can inhibit ET-1 formation may be different to those required for NO’s direct vasodilatory effects, but this has not been defined in detail. These interactions should be considered in all future studies to help explain the current inconsistencies between studies.

There are several ways in which NO could affect ET-1-mediated effects (Figure 1). This could be through the direct inhibition of constriction in response to released ET-1 acting at VSMCs. This is supported by the observation that NOS inhibition in vivo augments the constriction to endogenous ET-1, as assessed by blocking with ET receptor antagonists [42,43,44]. It is also possible that NO acts upstream by inhibiting the synthesis and release of ET-1. This could be via numerous pathways, including the inhibition of the transcription of preproET-1, translation, precursor conversion and exocytosis. A third potential mechanism was suggested by Goligorsky et al., whereby NO can diminish the duration of interaction between ET-1 and its receptors [93]. However, the underlying mechanisms have not been explored.

### 5.1. In Vitro

In cultured cells, NO has been shown to inhibit ET-1 transcription and release. The treatment of porcine aortic ECs with the NOS inhibitor, N(G)-monomethyl L-arginine (L-NMMA), potentiated the basal transcription and release of ET-1 [35], yet did not affect the thrombin-stimulated release of ET-1 [24]. In contrast, in HUVECs, the NO donor sodium nitroprusside (SNP) inhibited the transcription of both basal and hypoxia-induced ET-1, as measured by a nuclear run-off assay [34]. However, it was only in hypoxic conditions that SNP also reduced the release of ET-1. This suggests the effect of NO on ET-1 formation and release varies depending on the cell type and stimulus, and should be investigated further.

One study using HUVECs demonstrated that exposure to exogenous NO, at concentrations of 20 ppm or 80 ppm for 24 h, reduced intracellular levels of ET-1 as measured by ELISA compared to levels prior to NO exposure [36]. This was not coupled to a decrease in preproET-1 mRNA, suggesting NO might be inhibiting ET-1 at the point of translation, storage or release rather than transcriptionally. However, as they did not measure the release of ET-1 into the culture media, it is unclear if release was also affected.

### 5.2. Ex Vivo and In Vivo

In ex vivo preparations, the effect of NO is less clear. In isolated rat aortas, 35 min exposure to the NOS inhibitor, L-N(G)-Nitro arginine methyl ester (L-NAME) did not affect intracellular levels of ET-1 [14]. Similarly, in isolated porcine aortas, it was found that treatment with the NOS inhibitor, L-NMMA, for 4 h did not cause an increase in the basal release of ET-1, but did potentiate ET-1 release in arteries stimulated with thrombin [23,24]. This potentiation was also observed following exposure to methylene blue, which inhibits cGMP formation, and was reduced on treatment with the cGMP analogue, 8-bromo cGMP [23]. This suggests that downstream signalling via cGMP is important for NO to inhibit ET-1 release. While these results are consistent in aortas, mechanisms in other ex vivo vascular beds and the marked difference to cultured cells have yet to be explained.

At the level of whole organs, when measuring the coronary effluent levels of ET-1 in perfused rat hearts treated with L-NNA for 50 min, ET-1 levels positively correlated with elevated perfusion pressure and vascular leakage [40]. The elevation in pressure could be prevented with PD-142893, a non-selective ET receptor antagonist, but whether ET-1 release remained elevated was not assessed. Furthermore, following the application of S-nitroso-N-acetylpenicillamine (SNAP), an NO donor, ET-1 levels were decreased below basal levels and there was an increase in coronary flow [40]. However, the authors did not account for the effect of different flow rates on the concentration of ET-1 when collecting samples of coronary effluent. The latter suggests NO reduces ET-1 levels as the ET receptor antagonist had a similar effect and, when added in combination with L-NNA, PD-142893 largely attenuated post-ischaemic impairments. This also supports the suggestion that increased ET-1 levels lead to certain pathologies and that an imbalance in NO and ET-1 may cause this.

In perfused rat hearts treated with L-NNA for 15 min, there was no increase in ET-1 in the coronary effluent despite a reduction in coronary flow [94]. The L-NNA-mediated coronary artery constriction appeared due to ET-1, as it was prevented by either bosentan, an ET_A/B_ receptor antagonist, or BQ-123, an ET_A_ receptor antagonist. The authors theorised that NO normally suppressed basal ET-1 vasoconstriction, so the same concentration of ET-1 reduced flow in the presence of L-NNA.

In vivo experiments in rats have shown NO does not attenuate hypoxia-induced ET-1 release in pulmonary cells [41]. Exposure to 10% oxygen for 24 h lead to a significant increase in both ET-1 peptide and preproET-1 mRNA levels in lung tissue. When the rats inhaled 10 or 100 ppm NO, this increase in ET-1 expression in hypoxic lung tissue was not altered. The same study established the same pattern in cultured bovine pulmonary ECs, an example where in vitro and in vivo findings are matched.

It is evident that ET-1 and NO are in balance with each other within the healthy endothelium. However, it is still less clear whether NO has direct effects on ET-1 release or whether the balance is mainly maintained through NO-induced vasodilation acting in opposition to ET-1-induced vasoconstriction.

Overall, the complexity and variability of these findings underly the importance of establishing the mechanism for ET-1 storage and release in a particular tissue of interest under controlled conditions, before establishing changes in disease.

## 6. ET-1 in Disease

### 6.1. Cardiovascular Disease

As a potent vasoconstrictor, it is unsurprising that ET-1 has been implicated in many cardiovascular diseases (CVDs) and is commonly associated with endothelial dysfunction, which underlies disease.

In most cases, plasma and sometimes tissue levels of ET-1 have been found to be elevated in animal disease models and human patients (Table 2). However, there are still major gaps in our knowledge of what the role of ET-1, if any, is in these diseases.

#### 6.1.1. Hypertension

As a vasoconstrictor, perhaps the most obvious pathology would be elevated blood pressure (see Section 2.2.1). In hypertension, it is possible that an increase in plasma ET-1 precedes blood pressure changes and continues as the disease progresses. In 5–10-week-old spontaneously hypertensive rats (SHRs) (i.e., during the development of hypertension) the levels of ET-1 in the plasma and basal release of ET-1 from mesenteric arteries were increased from weeks 5–6 onwards compared with age-matched normotensive controls [95]. In SHRs, both the elevation in ET-1 plasma levels and release from mesenteric arteries increased with age (5–6 weeks vs. 9–10 weeks). It is not clear which cells released ET-1 nor whether ET-1 was a biomarker or an active contributor to hypertension, although, given its upregulation during disease development, it is possibly contributes to disease progression.

A later study, however, did not report a difference in plasma ET-1 between 6 week-old SHRs and age-matched controls [96]. However, at 16 weeks, plasma levels of ET-1 were significantly greater in SHRs. The reason for the discrepancy between studies has not been established. The EC ET-1 immunoreactive content was also similar or even slightly lower in the aorta and mesenteric artery extracts from SHRs compared to controls at 6 and 16 weeks. This may be explained by an increase in ET-1 release but not synthesis or storage. It would be of interest to establish whether the mRNA for *Edn1* is altered in the ECs of these arteries in the SHRs during the development of hypertension. Notably, ET-1 does not appear to solely underlie hypertension in SHRs, as ET receptor antagonists had no effect against the development [131] or maintenance of hypertension [132].

Evidence suggests ET-1 is more important in malignant hypertension. Plasma levels of ET-1 in 6- and 18-week-old rats were the same in SHRs and age-matched controls, unless they were each treated with deoxycorticosterone acetate (DOCA)-salt over 4–8 weeks to mimic malignant hypertension. ET-1 levels were increased in the SHRs but not the controls [97]. This was also accompanied by a further increase in systolic blood pressure. The increase in ET-1 was not directly due to acute hypertension as neither short-term infusion of phenylephrine nor angiotensin II caused an increase in ET-1 levels. Whether other mechanisms, such as reduced NO bioavailability, play a role has yet to be established. In contrast to SHRs, the role of ET-1 in DOCA-salt SHRs is supported by the treatment of these rats with ET receptor antagonists. Blood pressure was reduced [133,134] and changes to resistance arteries, such as a reduction in lumen diameter [133], were reported upon treatment with bosentan, however, not to control levels. Perhaps not unexpectedly, this suggests other factors may be contributing to disease alongside ET-1.

A potential mechanism for the release of ET-1 in hypertension, specifically preeclampsia, has been reported. When treating pregnant mice with IgG from women with preeclampsia, a significant increase in preproET-1 mRNA expression was induced [120]. An ET receptor blockade was able to significantly attenuate the pathologies associated with preeclampsia, including hypertension and renal damage, suggesting the direct involvement of ET-1 in pathology. This was further supported in vitro through the treatment of human placental villous explants, HUVECs and immortalised trophoblasts with IgG, which resulted in excess ET-1 secretion. The activation of the angiotensin II type 1 receptor was responsible for the increase in ET-1 as it could be blocked by losartan or an autoantibody-neutralising 7 amino acid epitope peptide able to block the receptor. It was suggested that the activation of the receptor resulted in an increase in TNF-α which, via IL-6, leads to an increase in ET-1. Interestingly, ET-1 may also contribute to high serum uric acid levels in preeclampsia which tends to precede the onset of the clinical manifestations used in diagnosis [135]. High uric acid levels are associated with adverse cardiovascular events and renal disease, and as such, ET-1 may influence disease progression.

Clearly many facets of hypertension remain unexplained and, while ET-1 may play a role, this requires further investigation.

#### 6.1.2. Atherosclerosis

Inflammation and immune cell activation are considered important in all stages of atherosclerosis. ET-1 has strong links to inflammatory disorders due to its ability to increase vascular permeability and the expression of inflammatory adhesion markers and cytokines (see Section 2.2.6).

Studies in animal models such as pigs [99] and mice [115], as well as humans with both early [100] and symptomatic atherosclerosis [98], have demonstrated the pathological role of ET-1 in atherosclerosis with an increased level of circulating ET-1 as well as an increase in tissue ET-1 in these patients. For example, Lerman et al. demonstrated ET-1 as an early participant and marker for coronary endothelial dysfunction in patients in the early stage of atherosclerosis [100]. Coronary arteries constricted rather than dilated to acetylcholine, suggesting EC damage, and the already elevated plasma ET-1 levels were further increased upon the administration of coronary acetylcholine. The authors proposed that acetylcholine may stimulate the release of ET-1 rather than NO in the region of the plaque, associated with the altered pathophysiological state [100]. It is also possible that ET-1 is released due to the reduced production of NO secondary to endothelial dysfunction, for example. ET-1 may then contribute to atherosclerosis through inflammation (see Section 2.2.6).

In patients with symptomatic atherosclerosis, there was a significant correlation between plasma ET-1 and the number of plaque sites. The immunohistochemistry of atherosclerotic vessels showed ET-like immunoreactivity in VSMCs and ECs [98]. A similar localisation of ET-1 has also been observed in the large epicardial coronary arteries of coronary artery disease patients undergoing cardiac transplantation [112]. ET-like immunoreactivity was associated with ECs in the artery lumen, microvascular ECs of the adventitia as well as SMCs present in the plaques. The presence of ET-1 in macrophages and intimal SMCs in atherosclerotic lesions has also been noted elsewhere [113]. Denser ET-1 labelling was observed in active lesions when compared to non-active lesions, suggesting a possible role in the disease, including the potential recruitment of immune cells to the plaques (see Section 2.2.6).

An increase in the aortic expression of ET-1 has also been reported in apolipoprotein E knockout (ApoE^−/−^) mice [115,116]. At lesions, the ET-1 label was localised to macrophage-foam cells as well as intimal and medial VSMCs. An increase in ET_A_ receptor expression was observed in medial SMCs but not macrophages, which suggests medial SMCs may be the target cells of ET-1 in this pathology [115]. In comparison, an immunohistochemical study of human aortas post-mortem demonstrated an increase in ET-1 and ET_B_ receptors in both unfoamy and foamy macrophages, as well as T cells and ET_B_ receptors in medial SMCs [114]. The differences in ET receptor type between the two studies could be species- or disease-progression-specific, but remain unexplained.

Furthermore, ET-1 can act as an independent predictor of coronary artery disease and other microvascular dysfunction. This could suggest it is directly involved in disease progression. For example, the plasma levels of big ET-1 have been shown to predict the severity of coronary artery calcification [102]. Coronary artery calcification is a direct marker of atherosclerosis and is significantly associated with all-cause mortality. In patients with stable typical or atypical chest pain, a score > 400 was independently predicted by a higher level of big ET-1. This also highlights the role of ET-1 in diagnostics.

In patients with recent myocardial infarction or those with underlying chest pain, plasma ET-1 levels have been shown to have a positive association with coronary artery disease status and the need for revascularisation by coronary artery bypass surgery [101], potentially suggesting the contribution of ET-1 to disease severity. The role of ET-1 in patients with angina, but without large coronary artery disease, has yet to be established.

Nevertheless, from all of the above studies, given the complexity of atherosclerosis, it is not possible to conclude if ET-1 is participating in the disease or is a disease marker. By using HUVECs, it has been shown that the overexpression of ET-1 resulted in an increase in mRNA levels of adhesion molecules such as intercellular adhesion molecule 1 (ICAM-1), and chemokines like monocyte chemoattractant protein-1. In a co-culture with either mouse or human macrophages, this resulted in the migration of the macrophages, a shift to M1-like phenotype and an increase in proinflammatory cytokines. This effect was markedly attenuated by an ET_A_ receptor antagonist, or followed by the inhibition of PKC, which the authors suggested was due to the activation of PKC-dependent ET-1 synthesis and release (see Section 4.1.1) caused by endoplasmic reticulum stress; however, they did not assess differences in ET-1 expression. Furthermore, in ApoE^−/−^ mice crossed with mice overexpressing ET-1, specifically in the vascular endothelium, there was a significant increase in plaque size and proinflammatory markers such as ICAM-1 and TNF-α, and a greater abundance of CD68 in the plaque, demonstrating increased macrophage infiltration [116]. This suggests ET-1 accelerates atherosclerosis disease progression via macrophage activation and recruitment to the plaque, and may provide a novel therapeutic target.

#### 6.1.3. Heart Failure

The role of ET-1 in heart failure (HF) is difficult to define, yet ET-1 levels may correlate with the survival rate of patients in HF. In rats that survive coronary artery ligation, there is increased mortality in the following months due to severe congestive HF [117]. In the left ventricle of these HF rats, the expression of both preproET-1 mRNA (as demonstrated by northern blot) and ET-1 peptide (visualised using immunocytochemistry and quantified using a sandwich-enzyme immunoassay performed on left ventricle homogenates) was increased. There was no difference in prepro-ET-1 mRNA levels in the kidneys between the controls and the HF rats, suggesting this was specific to cardiac tissue. The survival rate of these rats almost doubled when the rats were given a continuous infusion of BQ-123 over 12 weeks. BQ-123 administration also improved left ventricular dysfunction and prevented pathological ventricular remodelling, such as increased ventricular mass and an enlargement of the ventricular chamber. Fibrotic scarring in the left ventricle was still observed. It therefore appears that the continued administration of BQ-123 prevents the excessive myocardial hypertrophy which leads to HF.

Interestingly, it is possible that cardiac ischaemia triggers the local production of ET-1 in the macro- and microcirculation. In pigs subjected to 90 min ischaemia, there was an increase in plasma ET-1 which remained elevated on reperfusion [103]. Furthermore, when comparing ischaemic with non-ischaemic myocardium, a significant increase in ET-1 mRNA was detected in cardiomyocytes in the ischaemic region with little in vascular ECs. As this correlated with big ET-1, it is likely that the ET-1 is locally produced by these cells and is likely responsible for the maintained production of ET-1 on reperfusion.

The mechanism responsible for this increased ET-1 production is unknown. Peptides isolated from the blood of healthy subjects and post-myocardial infarction patients were added to human endothelial progenitor cells for 24 h. Those peptides isolated from the blood of post-myocardial infarction patients increased the release of ET-1 from these cells [136]. The identity of the peptides was not established, but clearly needs to be defined.

ET-1 may contribute to the pathology of HF through cardiomyocyte death. When simulating an ischaemic event on cultured neonatal rat myocytes, in as little as 30 min, ET-1 had a direct cytotoxic effect on cells with a dose-dependent increase in lactate dehydrogenase release (marker of cell damage) during ischaemia. This was not observed under non-ischaemic conditions [127]. Whether an ET receptor antagonist could prevent cell death was not explored.

Another potential mechanism by which ET-1 contributes to HF is by promoting fibrosis via ET receptors on cardiac fibroblasts [125,126,137]. Both receptor subtypes, ET_A_ [125] and ET_B_ [126], have been implicated separately. However, as fibrotic scarring was still reported in the left ventricle of HF rats treated with BQ-123 [117], it is possible that ET_B_ receptors are responsible for promoting fibrosis. The IL-6/STAT3/ET-1 pathway has been proposed to cause this fibrosis. SHRs treated with the lipid lowering drug, atorvastatin, had lowered systolic blood pressure and attenuated myocardial fibrosis. These were correlated with reduced plasma and cardiac tissue levels of ET-1 and IL-6 [124]. This attenuation was reversed in the inhibition of STAT3, a downstream regulator of both IL-6 and ET-1. Together, these pathways likely contribute to the pathological effects of ET-1 in the event of cardiac ischaemia resulting in HF.

It would appear, however, that ET-1 is beneficial in preventing HF in the short-term. In rats with HF following coronary artery ligation, the blocking of ET_A_ receptors with BQ-123 decreased both heart rate and myocardial contractility [128]. Similarly, in a mouse model of HF, ET-1 was shown to maintain normal heart function after chronic pressure overload [129]. In vascular EC-specific ET-1 deficient mice, hypertrophy was exacerbated compared to wild-type mice, while cardiac dysfunction was observed only in the knockout mice. It appears that ET-1 inhibits the pro-apoptotic TNF-α pathway as a blocker of this pathway, pentoxifylline, was able to prevent the pathological changes observed in the ET-1-deficient mice [129].

These studies suggest the upregulation of ET-1 release and its subsequent actions may maintain cardiac function in the short term, however, this becomes pathological in the longer-term. Potential therapeutic intervention could include either an appropriately timed ET receptor blockade or the regulation of ET-1 release.

#### 6.1.4. Air-Pollution-Induced Vascular Pathology

Given the rising levels of pollution worldwide and increase in pollution-associated deaths, ET-1, in the context of air-pollution-induced vascular dysfunction, has been explored recently. Vascular dysfunction caused by air pollution is characterised by systemic inflammation and endothelial dysfunction and, thus, provides a mechanistic link between air pollution and cardiovascular disease [138,139].

In children living in Mexico City where air pollution levels are high, a positive correlation between the number of hours spent outdoors and plasma ET-1 levels was found [104]. This could then result in pathological changes in the vasculature, although the longer-term effects of increased plasma ET-1 were not explored. Controlled human exposure experiments support this and suggest that acute exposure to air pollutants can increase ET-1 levels. An exposure to diesel exhaust at a dose of 100 μg/m^3^ was found to significantly increase the plasma concentrations of both ET-1 and matrix metalloproteinase-9 (MMP-9) [105]. Upon ET_A_ receptor activation, MMP-9 levels increase and, since MMP-9 can degrade the extracellular matrix [140], the dysregulation of MMP-9 activity may contribute to the pathological changes observed in the vasculature. MMP-9 has also been identified as a biomarker of the early stages of coronary heart disease [141]. Interestingly, a significant increase in aortic mRNA levels of ET-1 and MMP-9 have been observed in ApoE^−/−^ mice exposed to petrol emissions and carbon monoxide [121].

Air-pollution-induced vascular dysfunction is likely to become increasingly relevant and it is therefore important that we understand the role of ET-1 in this pathology.

#### 6.1.5. Diabetes-Associated Vascular Pathology

Diabetic patients are at a high risk of developing microvascular and macrovascular complications. This is likely, at least in part, due to the oxidative stress increased by both acute and chronic hyperglycaemia [142]. Oxidative stress alone has been shown to increase ET-1 transcription and release from HUVECs and bovine aortic ECs [143], and release from cultured rat pulmonary ECs [144]. Furthermore, in bovine retinal ECs and retinal pericytes, hyperglycaemia has also been shown to activate PKC via diacylglycerol (DAG) [145], leading to ET-1 production and release [145,146,147]. Furthermore, in bovine aortic ECs, insulin stimulated ET-1 production via the MAPK/ERK pathway [148]; this, coupled with the inhibition of eNOS activation via the downregulation of the PI3K/Akt pathway in insulin resistance [149,150], may further enhance levels of ET-1 and result in pathology. In patients with polycystic ovary syndrome, increased ET-1 levels are thought to contribute to the development of insulin resistance [151], which could be via the inhibition of glucose uptake and insulin-stimulated Akt phosphorylation [152]. It is possible that, in these diabetic patients, insulin and sulphonylurea treatment may exacerbate endothelial dysfunction. Epigenetic changes may also contribute towards increased levels of ET-1 in diabetes. Retinal microvascular cells cultured in high-glucose-containing media demonstrated hypomethylation in the proximal promoter and first exon region of the *Edn1* gene resulting in high expression of ET-1 mRNA [6].

ET-1 may be the mechanistic link between hyperinsulinemia and hypertension. In fructose-fed hypertensive rats, total ET-1 content in the mesenteric blood vessels was reported to be higher than in those of control animals [118]. The chronic blockade of ET_A/B_ receptors with bosentan resulted in a decrease in blood pressure without an accompanying alteration in ET-1 levels or plasma insulin, an effect not observed in control rats [118]. This suggests ET-1 is acting via its receptors to cause increased blood pressure in these rats, providing a potential mechanism by which ET-1 causes complications in diabetic patients.

An association between diabetes and cognitive impairment involving ET-1 has also been revealed. When microglia cells were cultured in high glucose to mimic diabetic conditions, an increase in the mRNA and protein levels of ET_A_ receptor and protein levels of ET_B_ receptor was observed. The microglia were also activated towards an M1-like phenotype with an increase in IL-17 and a decrease in CD206 (M2 marker) mRNA. M1-like microglia are thought to be proinflammatory with roles in tissue injury, therefore, this might result in neuronal degeneration. Upon exposure to LPS or hypoxia in diabetic conditions, an increase in ET-1 production via an increase in mRNA and secretion, which also resulted in the activation of the M1-like phenotype, were reported [122]. This demonstrates a mechanism parallel to atherosclerosis (see Section 6.1.2) involving ET-1 during diabetes resulting in pathologies within the brain.

### 6.2. Other Diseases

#### 6.2.1. Cancer

ET-1 is thought to progress tumours in colon, as well as ovarian, prostate, lung, bladder and breast cancers, via various mechanisms, including cell proliferation, resistance to apoptosis, angiogenesis, immune response modulation and metastasis, with evidence for the ET_A_ receptor activity playing a key role [72]. Indeed, ET receptor expression and activation of the ET-1 signalling pathway has been negatively correlated with patient outcome in several different cancers [153,154,155].

In the very aggressive, triple-negative breast cancer, the lactoferrin-mediated increase in breast cancer cell invasiveness is thought to be mediated at least in part by an increase in the expression and secretion of ET-1. The treatment of breast cancer cells with BQ-123 abolished the ability of ET-1 and lactoferrin to promote invasion [156]. The role of ET-1 in metastasis may be due to the ability of ET-1 to regulate proteinases such as the MMPs (see Section 6.1.4).

ET-1 may also prevent immune cell infiltration into tumours, thus aiding tumour progression and reducing the efficacy of immunotherapies for the treatment of cancer [123,130]. In tumour ECs from human ovarian cancer, an overexpression of the ET_B_ receptor was correlated with an absence of tumour-infiltrating lymphocytes and shorter patient survival time [123]. The importance of ET-1 and ET_B_ receptors in preventing this T cell homing was investigated by treating TNF-α-stimulated HUVECs with BQ-788. The treatment with BQ-788 led to increased T cell adhesion, an upregulation in ICAM-1 transcription and ICAM-1 clustering. The treatment of mouse cancer models also led to increased T cell homing in vivo. NO synthesis, stimulated by ET-1, appears to be important in this mechanism as L-NAME also restored T cell adhesion [123]. This is an example where NO augments the pathological effects of ET-1.

It is currently unclear if it is an increase in ET-1 release or an increase in the ET receptors on tumour cells, that is most influential in tumour progression. However, most of the clinical trials targeting the ET-1 signalling pathway have focused on receptor inhibition and many of these trials have been unsuccessful [72]. Therefore, it is possible that targeting the release of ET-1, rather than its signalling, is a new avenue to try in cancer therapy.

#### 6.2.2. Chronic Pain

ET-1 has also been implicated in chronic pain via its ability to interact directly with nociceptors. ET_A_ is expressed in the peripheral afferent nerve endings, nerve axons and nociceptor cell bodies in the dorsal root ganglion, suggesting it is made and transported to the nerve terminals [157]. ET-1 can also potentiate other algogens such as formalin [158].

Pain, both acute and chronic, is a clinical hallmark of sickle cell disease. Due to the role of ET-1 in pain, it has been implicated in this disease. In a humanised mouse model of sickle cell disease, levels of ET-1 mRNA and protein, and ET_A_ receptor protein expression, were increased in the dorsal root ganglia compared with controls, suggesting the upregulation of the ET-1 pathway. The inhibition or specific knockdown of the ET_A_ receptor in primary sensory neurons of the dorsal root ganglia reduced pain hypersensitivity, as assessed by paw withdrawal [119].

In patients suffering from rheumatoid arthritis and osteoarthritis, there are higher circulating levels of ET-1 compared to healthy subjects [107]. Furthermore, when comparing patients with active and inactive rheumatoid arthritis, it was found that patients with the former had higher plasma ET-1 [107], suggesting a role for ET-1 in disease severity.

#### 6.2.3. Asthma

As mentioned previously (see Section 2.2.3), ET-1 is also a potent bronchoconstrictor and is, therefore, implicated in asthma. Patients experiencing an acute asthmatic episode have increased bronchoalveolar lavage levels of ET-1 which decreased on management of the exacerbation [111]. Although this group did not report increased plasma levels of ET-1 in asthmatics, this has been shown previously with the severity of acute asthma being positively correlated with ET-1 plasma levels [106].

It is possible that asthmatic patients have an increased sensitivity to ET-1. Chalmers et al. described bronchial hyperactivity in response to inhaled ET-1 in asthmatic patients compared with healthy controls [159]. Although they were unable to provide a mechanism and previous research has suggested there is no difference in receptor expression between asthmatics and healthy volunteers [160], they did theorise that the damaged pulmonary epithelium was unable to mitigate the actions of ET-1, such as the clearance and enzymatic degradation of ET-1 or release of NO.

#### 6.2.4. COVID-19

Particularly relevant in recent years has been the involvement of ET-1 in the pathogenesis of COVID-19 infection.

Plasma levels of ET-1 have been correlated with disease severity. Healthy controls, asymptomatic and mild disease patients all had lower plasma ET-1 levels than patients requiring hospitalisation; as such, ET-1 plasma levels are a significant independent predictor of hospitalisation [108]. ET-1 levels were also significantly higher in patients suffering from complications such as acute kidney and myocardial injury, and death. Furthermore, patients with higher viral levels, which were associated with more severe disease and an increase in all-cause death, had higher ET-1 plasma levels than those patients with lower viral loads [110].

High plasma levels of ET-1 were also reported during acute infection, however, it was also reported that plasma levels were further elevated 3 months post-infection, alongside elevated levels of inflammatory cytokines, such as IL-6, and the activation of the coagulation cascade [109]. Interestingly, no macrovascular dysfunction was observed, as measured by the carotid artery diameter response to cold water hand immersion, however, this does not exclude dysfunction on the microvascular level. It is also unknown whether this chronically elevated level of ET-1 is associated with long-COVID.

ET-1 production and release can be upregulated by inflammatory factors, such as IFN-γ and IL-1β, during disease. As angiotensin converting enzyme 2 is downregulated in COVID-19, accumulating angiotensin-II can also cause the upregulation of ET-1 expression via AP-1 [161]. Thrombin is also elevated as part of the thromboembolic disorder in COVID-19 [162], which can also stimulate ET-1 (see Section 4.1). Furthermore, there is some suggestion of pulmonary arterial hypertension in COVID-19 patients, in which ET-1 is heavily implicated [161].

ET-1 may be linked to the adverse effects reported with the use of mRNA vaccines to target coronavirus via angiotensin-II-induced increase in ET-1 release via decreased expression of angiotensin-converting enzyme 2, much like in the clinical disease [163].

## 7. Conclusions

The vasoconstriction to ET-1 is important in maintaining basal vascular tone and can influence the ability to match blood supply to tissue metabolic demand. But ET-1 has also been implicated in a wide range of cardiovascular diseases, whether as a marker of the disease or by active involvement in the pathology. Despite the clear importance of ET-1 and its well-defined target receptors, what is (surprisingly) still unclear is how ET-1 is stored and released. This is partly due to the range of different experimental models, tissues and techniques used to probe these aspects of ET-biology.

This may also explain why, despite the widespread influence of ET-1 in disease, very few therapies targeting the peptide have been successfully developed and employed. Currently, the only clinical condition for which there is an approved treatment available that targets ET-1 is pulmonary arterial hypertension. However, it remains unclear as to what extent these antagonists modify disease progression. Their effects are very modest overall but are also associated with adverse side effects. Moving forward, it is clearly very important to define how ET-1 is stored and released, as the ability to control this process has the potential to utilise the positive and restrain the negative effects of ET-1 and address the associated pathologies in a safe, effective way.

## Figures and Tables

**Figure 1 ijms-24-11295-f001:**
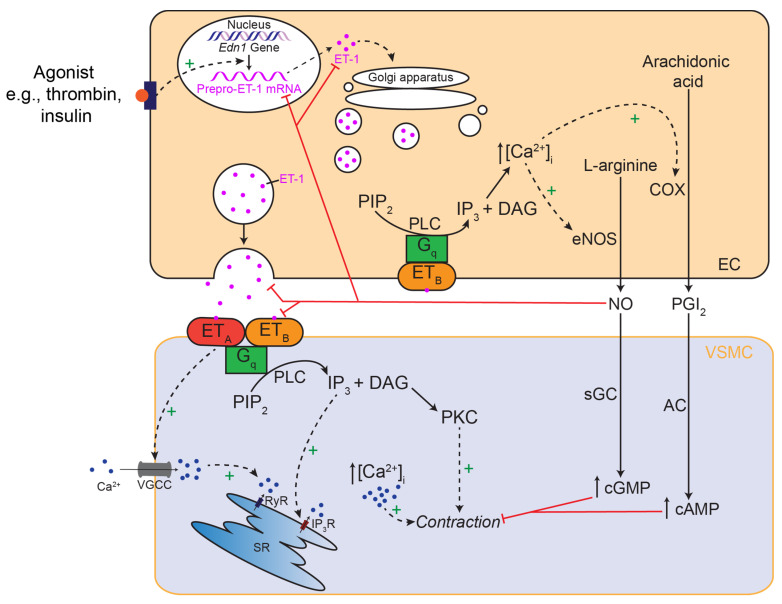
**Simplified schematic diagram of ET-1 signalling in the vasculature to cause vasoconstriction.** ET-1 is stimulated to be synthesised and/or released by the appropriate agonist from endothelial cells (ECs). ET-1 can then act via ET_A_ and ET_B_ receptors on the vascular smooth muscle cells (VSMCs). These receptors are G_q_-coupled which results in the hydrolysis of phosphatidylinositol biphosphate (PIP_2_) by phospholipase C (PLC) to inositol 1,4,5-triphosphate (IP_3_) and diacylglycerol (DAG). VSMC contraction is stimulated by the activation of protein kinase C (PKC) and an increase in [Ca^2+^]_i_ via inositol 1,4,5-triphosphate (IP_3_) acting on the sarcoplasmic reticulum (SR), activation of voltage gated calcium channels (VGCC) and calcium-induced calcium release via ryanodine receptors (RyR). ET-1 can also act on ET_B_ receptors on ECs, also via an increase in [Ca^2+^]_I_, to stimulate the release of vasodilators such as nitric oxide (NO) and prostaglandin I_2_ (PGI_2_) via endothelial nitric oxide synthase (eNOS) and cyclooxygenase (COX), respectively. NO and PGI_2_ can then diffuse into VSMCs to increase levels of cyclic GMP (cGMP) and cyclic AMP (cAMP) via soluble guanylyl cyclase (sGC) and adenylyl cyclase (AC), respectively, to inhibit contraction. NO may also have direct inhibitory effects on ET-1 via inhibition of transcription synthesis, release or ET-1-receptor interactions.

**Figure 2 ijms-24-11295-f002:**
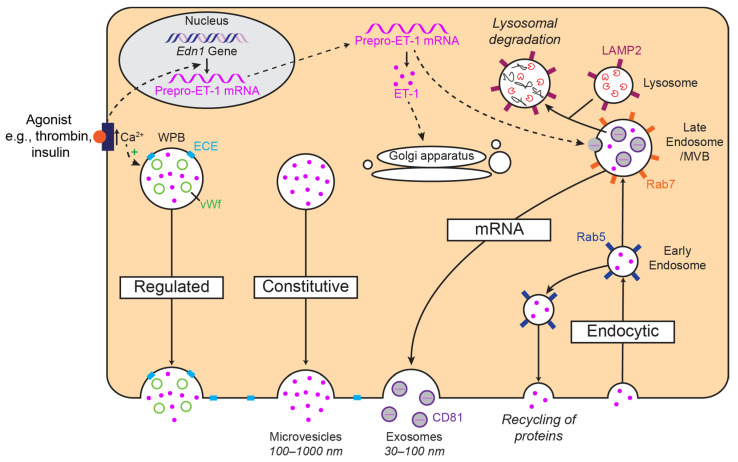
Simplified schematic diagrams of the proposed storage and release mechanisms of ET-1. ET-1 could be stored in Weibel–Palade bodies (WPB) and released via a regulated pathway or released via a constitutive pathway. ET-1 may also be recycled through the endocytic pathway. Through multivesicular bodies (MVB), there is also the possibility of storage and release of preproET-1 mRNA in exosomes. Several markers (e.g., Rab5) in this pathway have been highlighted as suggestions for future colocalisation studies.

**Table 1 ijms-24-11295-t001:** Summary of what is known about the ET-1 pathway from experiments performed in vitro (cultured cells such as porcine aortic ECs and HUVECs), ex vivo (intact tissue such as aortic rings) and in vivo (whole organism). ↑ is increased and ↓ is decreased.

		In Vitro	Ex Vivo	In Vivo
Transcription	AP-1	[5]		
	Epigenetic regulation	[6]		
Synthesis		[7,8]		[9]
Storage	Evidence for	[10]		[11]
	Evidence against	[12]	[13,14]	
	Constitutive secretory vesicles	[15]		
	WPBs	[16,17]	[18,19,20]	
	Endocytic pathway	[16]		
	No identified vesicle	[21]	[22]	
Release	Basal	[1,10,12,23,24,25,26,27,28,29,30,31,32,33,34,35,36]	[13]	
	Thrombin-induced	[12,23,24,26,27,29]	[13,14]	
	Angiotensin-II-induced	[25]		
	Vasopressin-induced	[25]		
	Neuropeptide-Y-induced	[33]		
	Lipopolysaccharide-induced	[32]		
	Cytokine-induced (TNF-α and IFN-γ)	[37]		
	Hypoxia	[34]		
	Calcium-dependent mechanism of release	[12,17,25,26]		
	PKC-dependent mechanism of release	[12,13,28]		
	PTK-dependent mechanism of release	[29]		
	cAMP-dependent inhibition	[31,32,33,37]	[38]	
	Shear stress-dependent inhibition	[28,39]		
Effects of nitric oxide	↓ Transcription/release	[10,28,34,35,36]	[40]	
	No effect on transcription/release	[24,41]	[14,23,24,41]	
	Inhibition of ET-1 actions			[42,43,44]
Signalling pathways		[45,46,47,48]		
Function	Vascular tone			[2,49]
	Immune response (e.g., ↑ vascular permeability, stimulate chemotaxis)	[50]	[51]	[52,53]
	Positive inotropy	[54]	[55]	[56]
	Kidney function (e.g., modulation of ion transport)	[57]	[58]	
	Airway function (e.g., bronchoconstriction)		[59]	[60]
	Gastrointestinal tract function (e.g., peristalsis)		[61,62]	

**Table 2 ijms-24-11295-t002:** Summary of changes associated with the ET-1 pathway in various diseases. ↑ is increased and ↓ is decreased.

Changes to ET-1 Pathway		Associated Pathology	Reference
Plasma ET-1 levels	↑	Early hypertension	[95]
		Hypertension	[95,96]
		Malignant hypertension	[96,97]
		Atherosclerosis	[98,99,100,101,102]
		Cardiac ischaemia	[103]
		Air pollution-induced vascular pathology	[104,105]
		Asthma	[106]
		Rheumatoid arthritis	[107]
		COVID-19	[108,109,110]
	No change	Early hypertension	[96]
		Hypertension	[97]
		Asthma	[111]
Tissue ET-1 levels	↑	Early hypertension	[95]
		Atherosclerosis	[99,112,113,114,115,116]
		Heart failure	[117]
		Diabetes	[118]
		Sickle cell disease	[119]
	No change	Early hypertension	[96]
ET-1 transcription	↑	Preeclampsia	[120]
		Heart failure	[117]
		Cardiac ischaemia	[103]
		Air-pollution-induced vascular pathology	[121]
		Diabetes	[6]
		Sickle cell disease	[119]
ET-1 release	↑	Early hypertension	[95]
		Preeclampsia	[120]
		Atherosclerosis	[100]
		Diabetes	[122]
		Asthma	[111]
ET receptor expression	↑ ET_A_	Atherosclerosis	[115]
		Diabetes	[122]
		Sickle cell disease	[119]
	↑ ET_B_	Atherosclerosis	[114]
		Diabetes	[122]
		Cancer	[123]
Mechanism of pathology involving ET-1	↑ Inflammation	Atherosclerosis	[116]
	Pathological tissue remodelling	Heart failure	[117]
	Fibrosis	Heart failure	[124,125,126]
	Cell death	Cardiac ischaemia	[127]
	Beneficial	Heart failure	[128,129]
	↓ Immune cell infiltration	Cancer	[123,130]

## Data Availability

No new data was created or analysed in this study. Data sharing is not applicable to this article.

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
