# Peer review of "Endothelin-1 in Health and Disease"

_ijms, 2023, doi:10.3390/ijms241411295_

Round 1
Reviewer 1 Report
The paper entitled “Endothelin-1 in Health and Disease” is well structured and presented. The paper review different aspects of endothelin 1, physiologically and related to pathology.
This work shows that some aspects related to endothelin-1 are not completely clarified, namely the storage and release of endothelin-1. This work can pave the way to in-depth study of these topics.
The review is a little bit long, but the topics developed justify the length of the document.
Suggestion of improvement:
Line 74 - GPCR receptor - the abbreviation in full for the first time.
Author Response
Dear Reviewer,
Thank-you for your comments, the minor change (GPCR) has been made.
Best wishes
Katie and Kim
Reviewer 2 Report
1. In the legend of Figure 1, "-Simplified" be better changed to "Simplified".
2. In the legend of Figure 2, "-Simplified" be better changed to "Simplified".
3. In line 446, "exposure[36]" be better changed to "exposure [36]".
4. In the legend of Table 1 "-Summary" be better changed to "Summary".
5. In the legend of Table 2 "-Summary" be better changed to "Summary".
Author Response
Dear Reviewer,
Thank-you for your minor comments, the changes to the text as indicated have been made.
Best wishes
Katie and Kim
Reviewer 3 Report
This paper is well written! Strategies aimed at the pathways influencing the synthesis and release of ET-1 pathway remain limited.
In circulation research, the forms of vascular arteries play an important role in endothelin -1 in Health and Disease. Blood flow in arteies should be take into consideration will become much better.
Author Response
Dear Reviewer,
Thank-you for your time in reviewing our paper and your positive comments. We have not made changes to the text.
Best wishes
Katie and Kim